# An Organizational Model of Online Learning in the Pandemic Period: Comparison with Traditional Face-to-Face Learning

Cristina Checa-Morales [1,2], Carmen De-Pablos-Heredero [1,3], Eduardo Díaz Ocampo [4], Yenny Guiselli Torres [5] and Antón García [2,*]

[1]   Business Administration (ADO), Applied Economics II and Foundations of Economic Analysis, Rey Juan Carlos University, 28032 Madrid, Spain; c.checa.2019@alumnos.urjc.es (C.C.-M.); carmen.depablos@urjc.es (C.D.-P.-H.)
[2]   Animal Science Department, University of Cordoba, Rabanales University Campus, 14071 Cordoba, Spain
[3]   Area of Business Economics, ESIC University, 28223 Madrid, Spain
[4]   Faculty of Business Sciences, Quevedo State Technical University, Carlos J. Arosemena Av., 120301 Quevedo, Ecuador; rector@uteq.edu.ec
[5]   Faculty of Animal and Biological Sciences, Quevedo State Technical University, Carlos J. Arosemena Av., 120301 Quevedo, Ecuador; ytorres@uteq.edu.ec
*   Correspondence: pa1gamaa@uco.es

**Abstract:** The COVID-19 pandemic has led to a paradigm shift in educational systems. During the lockdown, higher education became digital. This caused a change in communication within the educational ecosystem. Relational coordination (RC) is a communication and relationship model associated with the improvement of organizational results. Therefore, the objective of this research is to build an organizational model of online learning applied during the pandemic period and compare it with the previous face-to-face learning. A sample of 2774 students from two Ecuadorian universities was selected. A two-stage methodology was applied: First, an organizational model of online learning was built using multivariate methods. The RC model was linked to student satisfaction using generalized linear models (GLM). In the second stage, the organizational differences between the 2018 face-to-face and the 2020 online learning were identified. Finally, the online learning model was validated with external data. The components associated with a higher level of RC were institutional cooperation, institutional problem-solving, and administrative communication. Administrative communication lost importance in the online model. Significant differences between the satisfaction of the two models were not found. Nevertheless, since online learning was not associated with an improvement in satisfaction, the creation of a third educational model that combines the best practices of online and face-to-face learning in a hybrid system could be an alternative that improves the satisfaction of students.

**Keywords:** relational coordination; satisfaction; organizational model; higher education; online learning; COVID-19





## 1. Introduction

The COVID-19 pandemic forced a sudden change in teaching modalities [1]. Online methods were established to respect the imposed confinement and avoid contagion [2]. The change from traditional face-to-face learning to online classes involved a technological change and accelerated the use of digital technologies in students [3]. The roles of the administration and services staff, and of the teaching staff, were modified in an attempt to adapt to the reality of the pandemic. Besides, it affected the elements of student satisfaction [4]. However, before this change took place, there were already online and mixed systems implemented in universities worldwide [3]. Before the COVID-19 pandemic, online learning offered training options to specific student profiles, such as those that combined their training with work or family responsibilities [3]. During the pandemic period,

online learning was established immediately regardless of the technological capacity of universities and their staff [5,6]. The pandemic has led to a change in the relationships between the different actors in the higher education (HE) system. Therefore, this caused changes in the organizational model that could affect the quality of HE [1,5]. Numerous authors have studied this topic, offering different approaches to communication in HE in a context of confinement [1,5,7]. It is important to know how the change in education caused by the pandemic affected the organizational structure of the university and student satisfaction. It is also relevant to delve into the knowledge of the online learning that was adopted in confinement and check if it can improve the results of the universities and if it could be consolidated. It may be useful to extend it to other universities and improve the organizational structure in future educational systems.

### 1.1. Communication in HE in the Pandemic Period: Literature Review

During the confinement period, communication at the university was affected [1,5]. Given the need to limit physical contact, information and communication technologies (ICTs) have become a crucial element; so, digital literacy has been an essential element in online learning during the pandemic period [7,8]. According to Tejedor et al. [9] and Simon et al. [10], the learning scenario established by the pandemic highlighted the need to improve the digital skills of university staff. In this context, communication was the key element to provide students with personalized information and encourage their participation [11]. Sosa Díaz et al. [4] and Van-Der-Velde et al. [12] studied student satisfaction in the online classroom, concluding that communication skills through ICTs are necessary in the learning process both for accessing training content and to ensure optimal communication between the student and the other university worker's profiles—mainly lecturers and administrative staff. This opinion is shared by Harati et al. [13] and Schwanenberger et al. [14], who observed it both in students and in university administrative officers in online teaching. Obtaining timely feedback from lecturers in the evaluation process was crucial for students during online education in the pandemic period [15–17]. Furthermore, Flores et al. [5] and Prieto-Ballester et al. [18], showed that the level of digital literacy for online communication during confinement directly and positively affected students' well-being in online classes. In other words, HE requires quality organizational measures in the face of the change caused by COVID-19 [15,19,20].

### 1.2. RC and Improvement of Results

RC is an organizational model for task integration, where communication and relationship constitute the two main dimensions [21,22]. The communication dimension includes elements such as accurate and frequent communication aimed at solving problems [3,22–25]. The relationship dimension focuses on shared goals, knowledge, and mutual respect among all profiles of an organization [3,22–25]. The RC model has been applied in different sectors such as healthcare [22], banking [26], or airlines [21]. In addition, it has been studied in HE, showing that an increase in RC leads to an improvement in the results [3,23,24,27–29].

In the context of COVID-19, Sulmonte et al. [30] showed how the Quality and Safety team was able to harness skills in relational coordination and process improvement to respond to rapidly changing needs in a flexible and effective manner. In the field of HE, it is interesting to see if the drastic change in the educational paradigm (from face-to-face to online learning) has had an effect on student satisfaction [31,32]. Due to the unpredictability of the change in the global situation [1,2], there is a lack of studies comparing organizational models of communication in HE before the pandemic and in COVID-19 pandemic periods.

Therefore, the objective of this research was to build an organizational model of online learning applied at Quevedo State Technical University (UTEQ) during the pandemic period (2020) and compare it with the face-to-face (2018) learning at the same university.

Firstly, an organizational model of online learning was built in the period of the COVID-19 pandemic (May to September of 2020) UTEQ in Ecuador. The organizational model was linked to student satisfaction. Secondly, the organizational model obtained was

compared with the face-to-face model prior to the pandemic developed during 2018 at UTEQ. Finally, the organizational model was validated with external data from the State University of Bolivar (UEB) in Ecuador, during the period from May to September 2020.

To achieve the objective of this work, three research questions were established: (RQ1) What organizational changes have been experienced in learning as a result of the pandemic?; (RQ2) were differences observed between the satisfaction of face-to-face and online in pandemic period learning?; (RQ3) Can the best organizational practices of the model be extended to other universities?

This research will help to determine whether there were organizational differences between the online learning established during the pandemic and the previous face-to-face learning. In addition, it will allow knowing if there were changes in student satisfaction and will enable the proposal of specific organizational practices for future educational systems.

After this introduction, the second section presents the case of UTEQ; the third describes the materials and applied methods; the fourth shows the results; the fifth section exposes the discussion; the sixth, the limitations of the work; and, finally, the seventh section contains the conclusions.

## 2. The case of UTEQ in Ecuador

According to The World Bank [33], the COVID-19 health crisis triggered a recession, meaning that Ecuador needed to make its public practices more efficient and improve the quality of its educational system. These efforts require evidence-based decision-making and better management of public resources. To this end, The World Bank approved the Country Partnership Framework (CPF) 2019–2023 with Ecuador. Among its main objectives, the development of human capital and improving institutional sustainability are being promoted.

The ranking of the Council for Evaluation, Accreditation, and Quality Assurance of Higher Education (CEACEES) was considered to observe the category of the university. This is the entity in charge of accrediting the position of each university in the internal ranking of universities in Ecuador; the categories go from "A" to "D" on a decreasing scale and are determined for a period of five years [34].

Data from Quevedo State Technical University (UTEQ) were studied in this work. UTEQ is a university located in the coast zone of Ecuador, in Quevedo. It corresponds to category "B" in the CEACEES ranking [34]. This university offers 33 undergraduate and 15 postgraduate degrees [35]. The UTEQ online learning data collected in 2020 (UTEQ_2020) were used to perform the main analyses of this study and were compared with the UTEQ results of face-to-face learning in 2018 (UTEQ_2018) [23].

### 2.1. HE Prepandemic Context: Face-to-Face Learning at UTEQ_2018

In Checa et al. [23] the organizational model in face-to-face higher learning at UTEQ (2018), by using the RC, was widely described. It was linked to student satisfaction as a metric of university quality. The most relevant components were administrative communication, student leadership, lecturer cooperation, classmates' coordination, classroom communication, and "autonomy". Altogether, these explained 66.23% of the variance.

The most influential element in satisfaction was administrative communication. The students considered administrative officers as a reliable source of information. It was recommended to simplify administrative processes and improve the quality of information through accurate and frequent communication, and the reduction of response times aimed at problem-solving. Student leadership showed that student representatives were associated with a higher level of satisfaction. They were presented as a method to reduce the asymmetry of information between the student and the rest of the university profiles, and a way to increase the value of the human capital of the students. In addition, lecturer cooperation and classmates' coordination were also elements associated with student satisfaction, although to a lesser extent. Finally, the least relevant element was "autonomy", which showed the lack of willingness of the students to solve problems on their own. The

research reported that an improvement in the two RC dimensions between students and the rest of the university profiles, the level of satisfaction, and the sustainability of the organization improve.

### *2.2. HE in Pandemic Period: Online Learning at UTEQ_2020 and UEB_2020*

The COVID-19 pandemic caused the closure of higher education institutions unexpectedly. The move from face-to-face to online learning was forced to mitigate the spread of the virus. At UTEQ, this readjustment affected all degrees and subjects, both theoretical and practical. The university used the Microsoft Teams tool and Moodle as the digital platforms selected by the government for public institutions to deliver online teaching. For communication with administrative officers, the students used e-mail. However, difficulties related to digital skills and access to the internet or computing devices, as well as delays in response time, were observed [36]. This caused changes in the communication between the student and the rest of the university profiles.

UEB is a university located in Guaranda, a zone in the highlands of Ecuador, 140 km away from UTEQ. According to the CEACEES ranking, it corresponds to category "C" [34]. The UEB offers 20 undergraduate and 9 postgraduate degrees [37]. UEB was made up of 5000 students, compared to the 10,000 students of UTEQ. In the pandemic period, an online learning process similar to that of the UTEQ was developed, given the state nature of educational practices in the face of the COVID-19 crisis. The study of UEB data made external validation possible of the model obtained in online learning (UTEQ_2020) [38].

### 3. Materials and Methods

In the first stage, an organizational model was built for online learning (UTEQ_2020), through the development of an organizational typology. Principal component analysis (PCA) and cluster analysis were performed. Subsequently, the model obtained was related to student satisfaction through the use of generalized linear models (GLM). In a second stage, to deepen the organizational differences between face-to-face learning (UTEQ_2018) and online learning (UTEQ_2020), both models were compared. In a third stage, online organizational learning (UTEQ_2020) was validated with data from the online learning model of the State University of Bolivar (UEB_2020). The methodological stages are shown in Figure 1.

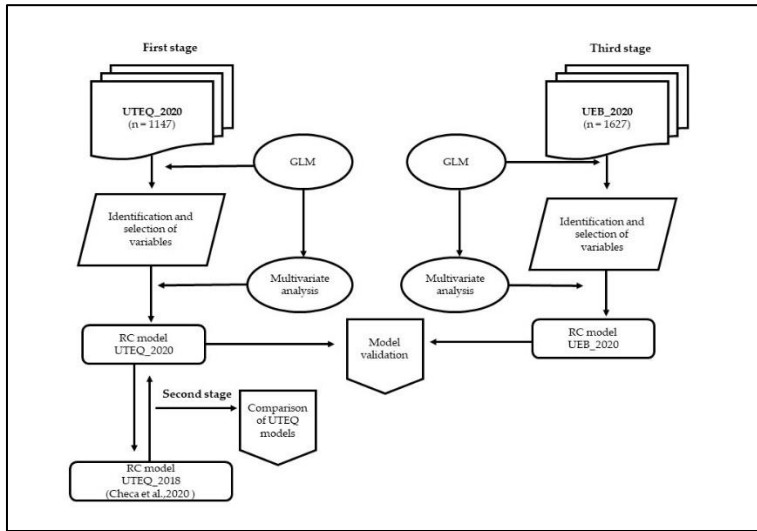

**Figure 1.** Methodology stages [23].

### *3.1. HE in Pandemic Period: Online Learning at UTEQ_2020 and UEB_2020*

The participants were students enrolled in undergraduate and postgraduate degrees in the following fields of knowledge: social sciences, humanities, engineering, and health sci-

ences. More than 3,000 surveys were collected between the months of May and September in 2020 at UTEQ and UEB. The surveys were distributed online through the Google Forms application. The answers of the students that were incomplete and those in which logical inconsistencies were observed were eliminated in the database cleaning process [39]. A stratified random sample composed of 1,147 participants from online learning (UTEQ_2020) was used to determine the main model. Finally, another stratified random sample composed of 1,627 participants from online learning (UEB_2020) was used to validate the model of online learning (UTEQ_2020).

*3.2. Relational Coordination Survey*

The students were asked through an online questionnaire composed of 4 questions on sociodemographic data (age, gender, course, and field of knowledge), 23 relational coordination items (11 communication and 12 relationship), and 6 items on satisfaction [21–23,25]. The original survey questions were asked in Spanish; these are shown with their English translations in Table 1. The questions in the survey were referred to all university profiles: lecturers, administrative officers, classmates, student representatives. The "myself" index was considered as a control variable [23]. In order to answer the survey questions obtaining metric variables, a Likert scale from 1 to 5 was used. Each equidistant point on the visual scale was associated with a level of student response [40]. The reliability of the questionnaire was verified using Cronbach's alpha, with values greater than 0.7 considered acceptable to confirm internal consistency: communication dimension (0.851), relationship dimension (0.938), and satisfaction (0.936) (Table 1). The complete survey (available as supplementary material) showed a Cronbach's alpha of 0.957 [41].

To measure satisfaction, variables 14 to 29 of the survey were used and a satisfaction indicator (SATISTotal) was built. The validity of the indicator was verified from the measures of central tendency (mean, median, and mode), dispersion (standard deviation and coefficient of variation), and asymmetry (kurtosis). Once SATISTotal was obtained, its main statistical descriptors (trend, dispersion, and asymmetry) were observed and a dichotomous satisfaction variable was determined for each university with two possible values: low satisfaction (LS) and high satisfaction (HS). A level of 19 points differentiated the two levels of satisfaction in UTEQ and a level of 20 points in UEB (Figure S1) [3,23–25].

Subsequently, a hierarchical cluster analysis was performed from the variables with higher variance values in the PCA. The grouping was carried out considering the intra and intergroup variance, joining the most akin cases to each other and different from the others, applying the Ward method. Besides, the Euclidean distance was used to check the degree of dissimilarity between the cases [47]. Once the groups were obtained, the analysis of variance (ANOVA) and the Student–Newman–Keuls method were applied to find significant differences between the sample means [48].

To answer RQ2 and verify the satisfaction levels of the obtained model, generalized linear models (GLM) were used. This analysis allowed determining which pairs of means show significant differences and analyzing variables with nonconstant variances and non-normal error [49]. GLM summarizes a similar group of regression methods, which were previously applied individually. A value of $p < 0.001$ was used as the level of significance. All statistical analyses were performed with Statgraphics Centurion XVI.I software.

In a second stage, the organizational typologies of face-to-face learning (UTEQ_2018) and online learning (UTEQ_2020) were compared; thereby, the differences in the resulting components were highlighted as well as the positions of the components in each model. In order to validate the resulting model and verify RQ3, the organizational model of online learning at UTEQ (UTEQ_2020) was compared with that of online learning at UEB (UEB_2020). Both were obtained in the period of the COVID-19 pandemic. This comparison allowed us to extend the model to other institutions [50].

**Table 1.** Relational coordination and satisfaction variables.

| Dimension | α Cronbach | Code | Question/Variable |
|---|---|---|---|
| COMMUNICATION | 0.851 | ACCURATE COMMUNICATION (Do the people who belong to these areas have the need to offer you information at certain times?) with | |
| | | 1.$\text{ACCU}_{\text{Admin}}$ | administrative officers |
| | | 2.$\text{ACCU}_{\text{Lect}}$ | lecturers |
| | | 3.$\text{ACCU}_{\text{Class}}$ | classmates |
| | | FREQUENT COMMUNICATION (Do people who belong to the following work areas communicate with you frequently?) with | |
| | | 4.$\text{FREQ}_{\text{Admin}}$ | administrative officers |
| | | 5.$\text{FREQ}_{\text{Lect}}$ | lecturers |
| | | 6.$\text{FREQ}_{\text{Class}}$ | classmates |
| | | SOLVING PROBLEM COMMUNICATION (When any type of problem appears (study, logistics, documentation . . . ), how much did the following profiles help you to solve your problem?) with | |
| | | 7.$\text{SOLPRO}_{\text{Myself}}$ | myself |
| | | 8.$\text{SOLPRO}_{\text{Lect}}$ | lecturers |
| | | 9.$\text{SOLPRO}_{\text{Repres}}$ | students' representatives |
| | | 10.$\text{SOLPRO}_{\text{Admin}}$ | administrative officers |
| | | 11.$\text{SOLPRO}_{\text{Class}}$ | classmates |
| RELATIONSHIP | 0.938 | SHARED KNOWLEDGE (How well do the following profiles know about your role in the university and the problems that arise?) with | |
| | | 12.$\text{SKNOW}_{\text{Lect}}$ | lecturers |
| | | 13.$\text{SKNOW}_{\text{Repres}}$ | students' representatives |
| | | 14.$\text{SKNOW}_{\text{Admin}}$ | administrative officers |
| | | 15.$\text{SKNOW}_{\text{Class}}$ | classmates |
| | | MUTUAL RESPECT (How much do the following profiles respect your role at the university?) with | |
| | | 16.$\text{RESPE}_{\text{Lect}}$ | lectures |
| | | 17.$\text{RESPE}_{\text{Repres}}$ | students' representatives |
| | | 18.$\text{RESPE}_{\text{Admin}}$ | administrative officers |
| | | 19.$\text{RESPE}_{\text{Class}}$ | classmates |
| | | SHARED GOALS (How well do the following profiles share your goals at the university?) with | |
| | | 20.$\text{SHARGOAL}_{\text{Lect}}$ | lecturers |
| | | 21.$\text{SHARGOAL}_{\text{Repres}}$ | students' representatives |
| | | 22.$\text{SHARGOAL}_{\text{Admin}}$ | administrative officers |
| | | 23.$\text{SHARGOAL}_{\text{Class}}$ | classmates |
| SATISFACTION | 0.936 | STUDENT SATISFACTION (Indicate your degree of satisfaction with the following profiles) with | |
| | | 24.$\text{SATIS}_{\text{Lect}}$ | lectures |
| | | 25.$\text{SATIS}_{\text{Represent}}$ | students' representatives |
| | | 26.$\text{SATIS}_{\text{Admin}}$ | administrative officers |
| | | 27.$\text{SATIS}_{\text{Materials}}$ | materials |
| | | 28.$\text{SATIS}_{\text{Communic}}$ | communication channels |
| | | 29.$\text{SATIS}_{\text{Contents}}$ | training contents |

## 4. Results

The main sociodemographic data of each sample are shown in Table 2. The majority of students surveyed in 2020 were female and younger than 25 years old. First- and second-year students and postgraduate students predominated in UTEQ_2020 and UEB_2020. In the 2020 samples, students from all university courses were considered, since the online modality during confinement affected all students. Most of the online learning (UTEQ_2020) students belonged to the engineering field of knowledge, due to the predominantly agrarian nature of the university. The distribution of the sample by field of knowledge in face-to-face learning (UTEQ_2018) confirmed this point. In UEB_2020, most of the students belonged to the social sciences, engineering, and health sciences for the most part, which confirmed

the heterogeneity of the sample. The description of the UTEQ_2018 sample is derived from the work of Checa et al. [23].

**Table 2.** Sociodemographic distribution of the samples (%).

| | | Age | | Gender | | Academic Year | | | | | | | Field of Knowledge | | | |
| | | <25 | >25 | Male | Female | 1st | 2nd | 3rd | 4th | 5th | 6th | 7th | Social sciences | Humanities | Engineering | Health sciences |
|---|---|---|---|---|---|---|---|---|---|---|---|---|---|---|---|---|
| UTEQ_2018 | 3200 | 43.36 | 56.64 | 38.71 | 61.29 | – | – | 45.94 | 42.98 | 11.08 | – | – | 37.25 | | 55.36 | 7.39 |
| UTEQ_2020 | 1147 | 88.40 | 11.60 | 40.71 | 59.29 | 19.70 | 13.51 | 10.46 | 12.29 | 13.86 | 7.67 | 22.49 | – | – | 100 | – |
| UEB_2020 | 1627 | 83.90 | 16.10 | 40.69 | 59.31 | 1.60 | 18.87 | 17.95 | 17.15 | 16.04 | 9.04 | 19.36 | 37.19 | 5.84 | 29.19 | 27.78 |

### 4.1. Organizational Model in Online Learning (UTEQ_2020)

#### 4.1.1. Organizational Typology of Online Learning (UTEQ_2020)

The components obtained in the online learning model (UTEQ_2020) are shown in Table 3. Five factors explained 70.51% of the variance. The first component explained more than 44% of the variance (Table 3). It displayed the highest values in variables such as mutual respect and shared goals related to all university profiles. "Institutional cooperation" was the name of the component. The second one justified 8.42% of the variance. The predominant variables were communication for problem solving and shared knowledge. The related profiles were lecturers, student representatives and administrative officers. This component was "institutional problem solving". The third component was "administrative communication". It explained 7.50% of the variance and was composed of the variables of accurate and frequent communication, related to lecturers and administrative officers. The fourth factor represented 5.27% of the variance. The variables of accurate and frequent communication prevailed, referring to the classmates profile. This factor constituted "classmates communication". Finally, the fifth factor explained the 4.65% of variance and was linked to problem-solving communication with "myself" profile. This component was called "autonomy".

**Table 3.** Principal components (PC) of online education (UTEQ_2020).

| Items | Loading | Eigenvalues | Explained Variance (%) | $\alpha$ Cronbach | PC |
|---|---|---|---|---|---|
| 16.RESPE$_{Lect}$ | 0.755 | 10.34 | 44.96 | 0.950 | 1 |
| 17.RESPE$_{Repres}$ | 0.791 | | | | |
| 18.RESPE$_{Admin}$ | 0.816 | | | | |
| 19.RESPE$_{Class}$ | 0.796 | | | | |
| 20.SHARAGOAL$_{Lect}$ | 0.809 | | | | |
| 21.SHARAGOAL$_{Repres}$ | 0.811 | | | | |
| 22.SHARAGOAL$_{Admin}$ | 0.791 | | | | |
| 23.SHARAGOAL$_{Class}$ | 0.790 | | | | |
| 8.SOLPRO$_{Lect}$ | 0.660 | 1.94 | 8.42 | 0.819 | 2 |
| 9.SOLPRO$_{Repres}$ | 0.763 | | | | |
| 10.SOLPRO$_{Admin}$ | 0.715 | | | | |
| 14.SKNOW$_{Admin}$ | 0.689 | | | | |
| 1.ACCU$_{Admin}$ | 0.730 | 1.73 | 7.50 | 0.760 | 3 |
| 2.ACCU$_{Lect}$ | 0.744 | | | | |
| 5.FREQ$_{Lect}$ | 0.653 | | | | |
| 3ACCU$_{Class}$ | 0.748 | 1.21 | 5.27 | 0.819 | 4 |
| 6.FREQ$_{Class}$ | 0.815 | | | | |
| 7.SOLPRO$_{Myself}$ | 0.897 | 1.00 | 4.65 | - | 5 |

Figure 2 represents the results of the cluster analysis of online learning (UTEQ_2020) with Ward's method, based on Euclidean distances. The centroids of each type of organization are shown in Table 4. The second group, with the highest levels of RC, grouped 45.51% of students and presented high figures in the five components ($p$-value < 0.05). The first

group, with medium RC, justified 39.49% of students and central figures in the centroids (*p*-value < 0.05). The third group, with lower RC levels, bunched 15.00% of students and showed negative figures in all components (*p*-value < 0.05).

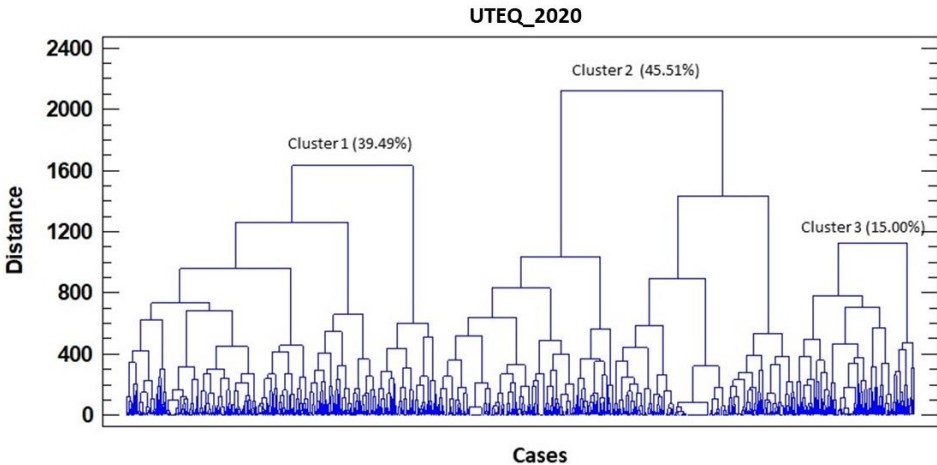

**Figure 2.** Relational coordination clusters of online education (UTEQ_2020).

**Table 4.** Centroids for each cluster of online education (UTEQ_2020).

| Components | PC [1] | Cluster 1 | Cluster 2 | Cluster 3 |
|---|---|---|---|---|
| Institutional cooperation | 1 | −3.907 [b] | 6.512 [c] | −9.476 [a] |
| Institutional problem-solving | 2 | −2.621 [b] | 4.537 [c] | −6.865 [a] |
| Administrative communication | 3 | −1.590 [b] | 3.553 [c] | −6.596 [a] |
| Classmates' communication | 4 | −1.037 [b] | 2.984 [c] | −6.325 [a] |
| Autonomy | 5 | −0.003 [b] | 0.504 [c] | −1.522 [a] |

[1] Principal component. [a,b,c] Within row, averages with different superscripts differ significantly, *p*-value < 0.001.

4.1.2. Satisfaction Level of Online Learning (UTEQ_2020)

The satisfaction rating in online learning (UTEQ_2020) was $21.27 \pm 0.15$, with a coefficient of variation of 24.43%. The data from this university showed positive Fisher asymmetry values and did not fit the normal distribution (Figure S1). The behavior of the satisfaction variable is represented in Figure 3. An increase in RC in the first two components managed to reach greater rates of satisfaction (*p*-value < 0.001).

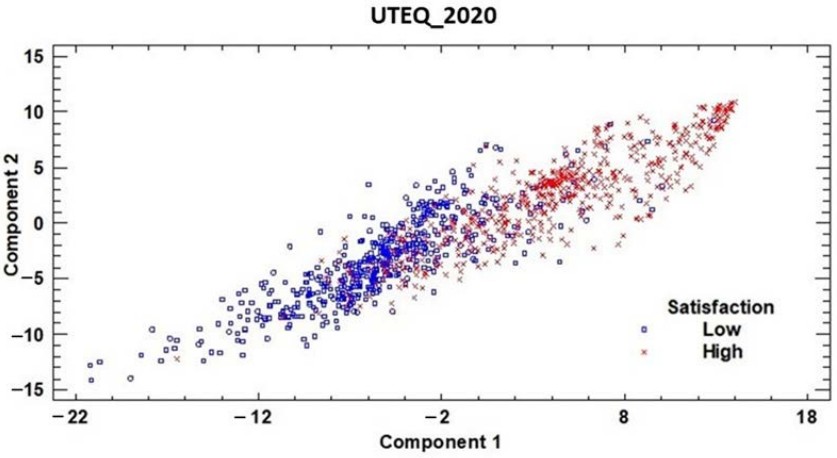

**Figure 3.** Dispersion of the satisfaction variable in online education (UTEQ_2020).

The relationship between RC and satisfaction (SASTISTotal) is shown in Figure 4. GLM results showed a significant linkage amidst satisfaction and cluster, with a confidence rate of 99%. Likewise, Duncan test for comparison of mean showed the existence of significant differences among clusters. In online learning (UTEQ_2020), the second cluster obtained the highest satisfaction values (*p*-value < 0.001). As for the lowest levels of satisfaction, these were observed in the third cluster. The satisfaction indicator showed differences between each group regarding its density function. This is shown in Figure S1 as supplementary material.

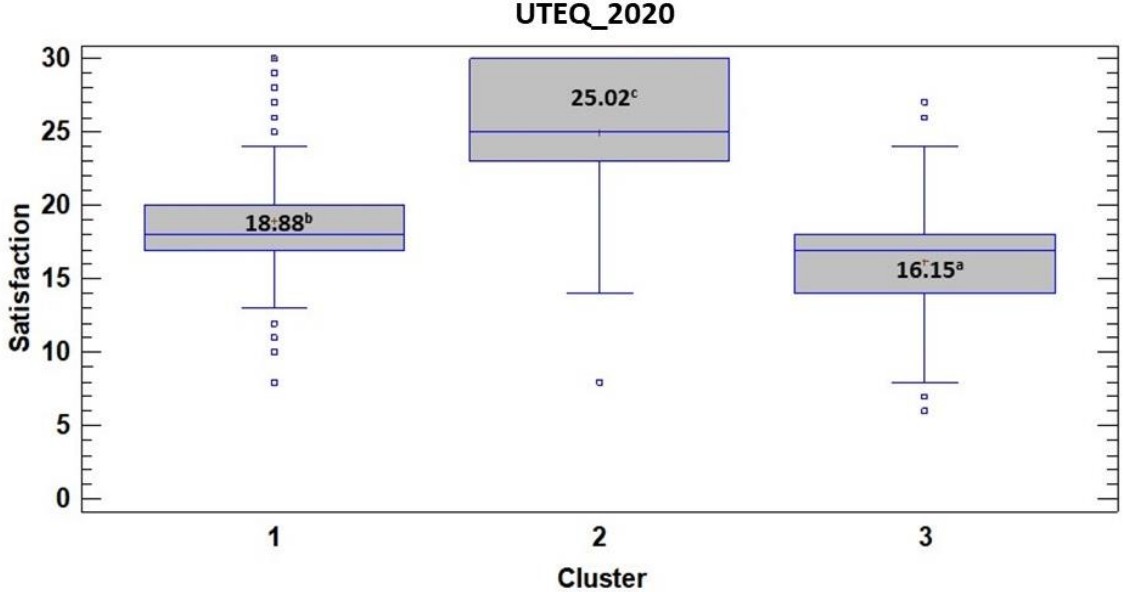

**Figure 4.** Total satisfaction in each relational coordination model of online education (UTEQ_2020). Means [a,b,c] with different superscripts differ significantly, *p*-value < 0.001.

### 4.2. Comparison between Face-to-Face (UTEQ_2018) and Online (UTEQ_2020) Learning

The main components in the typologies built for face-to-face (UTEQ_2018) [23] and online learning (UTEQ_2020) are shown in Table 5. Among the components with the highest explained variance (>60%), administrative communication and institutional cooperation were found in both models. Significant differences in satisfaction between face-to-face and online learning were not found (Table 6).

**Table 5.** Comparison of the components between face-to-face (UTEQ_2018) and online (UTEQ_2020) education.

| Face-to-Face Education | | | Online Education | | |
|---|---|---|---|---|---|
| Component Name | PC | Explained Variance (%) | Component Name | PC | Explained Variance (%) |
| Administrative communication | 1 | 36.13 | Institutional cooperation | 1 | 44.96 |
| Student's leadership | 2 | 8.58 | Institutional problem-solving | 2 | 8.42 |
| Lecturer cooperation | 3 | 7.25 | Administrative communication | 3 | 7.50 |
| Classmates' coordination | 4 | 5.26 | Classmates' communication | 4 | 5.27 |
| Classroom communication | 5 | 4.59 | Autonomy | 5 | 4.56 |
| Autonomy | 6 | 4.42 | — | — | — |

Administrative communication was the most relevant component in face-to-face learning. The variables' accurate, frequent, and problem-solving communication and shared knowledge with administrative officers made up this component. In online learning, the

predominant component was institutional communication. Mutual respect and shared goals with all profiles (classmates, lecturers, administrative officers, and student representatives) were the included variables.

**Table 6.** Satisfaction in face-to-face UTEQ_2018 and online UTEQ_2020 education.

| Face-to-Face Education | Online Education | *p*-Value |
|---|---|---|
| 23.35 ± 0.08 | 21.27 ± 0.15 | ns |

ns = not significantly different.

The components were rotated. The administrative communication component went from being the first in face-to-face learning to the third in online learning. The institutional cooperation component in online learning was similar to the lecturer cooperation in face-to-face learning. It collected the mutual respect and shared goals variables related to lecturers. In the rest of the components, the relationships between classmates in face-to-face learning (components 4 and 5) were combined in classmates communication (component 4) in online learning. Finally, the autonomy of the student formed by the problem-solving communication was not reliable in either of the two models.

*4.3. Model Validation*

The online model between both universities (UTQ_2020 and UEB_2020) is compared in Table 7. The components with the highest factor loading were similar in both cases. The factorial composition of the variance explained in the rest of the components was similar in both universities, except for the fifth component, "autonomy", in UTEQ_2020. This component was removed as it was not reliable according to Cronbach's alpha. The typologies built for UTEQ_2020 and UEB_2020 showed a similar factorial structure and grouping in cluster analyses. Significant differences between the satisfaction of UTEQ_2020 and UEB_2020 were not found (Table 8). Online learning UEB_2020 data analyses are shown in the supplementary material.

**Table 7.** Comparison between the components of online education in UTEQ_2020 and UEB_2020.

| UTEQ_2020 | | | UEB_2020 | | |
|---|---|---|---|---|---|
| Component Name | PC | Explained Variance (%) | Component Name | PC | Explained Variance (%) |
| Institutional cooperation | 1 | 44.96 | Institutional cooperation | 1 | 44.72 |
| Institutional problem-solving | 2 | 8.42 | Institutional problem-solving | 2 | 7.49 |
| Administrative communication | 3 | 7.50 | Administrative communication | 3 | 6.97 |
| Classmates' communication | 4 | 5.27 | Classmates' communication | 4 | 4.95 |

ns = not significantly different.

**Table 8.** Satisfaction in online education in UTEQ_2020 and UEB_2020.

| UTEQ_2020 | UEB_2020 | *p*-Value |
|---|---|---|
| 21.27 ± 0.15 | 19.76 ± 0.13 | ns |

ns = not significantly different.

## 5. Discussion

An organizational model for online learning at UTEQ in 2020 was built. The organizational models in face-to-face and online learning were different. In face-to-face learning before the pandemic, students mainly solved their administrative problems in person with administrative officers, since the response rate was higher than in telematic inquiries [51]. With the appearance of the pandemic, confinement forced people to work from home [20]. The administration staff did not have precision technological means in their homes, nor did

they have sufficient organization to provide student attention synchronously [36]. This, together with the increased influx of student queries on administrative procedures affected by the pandemic, caused delays in responses. Regarding the resolution of academic problems, before the pandemic, students turned to student representatives in face-to-face learning to solve academic and organizational problems [3,22–25]. In online learning, this profile had the same problem as the administrative officers: lack of adequate technological means in their homes and a higher rate of inquiries than was usual in face-to-face learning [6].

Therefore, the main organizational change was observed: while in face-to-face learning, the students solved the administrative and academic problems with the profiles responsible, respectively, in the online learning the students asked the lecturers directly. This was because the only synchronous contact in education was between student and lecturer, due to online classes. The lecturers made a great effort and reacted to the pandemic by implementing online classes with public and own equipment to respond to the pandemic and keep their communication channels (Microsoft Teams and Moodle) open. Consequently, in online learning, lecturers assumed the role of solving problems through synchronous and asynchronous communication during the class period.

In the comparison between the online and face-to-face model, students showed the same level of satisfaction in face-to-face and online learning. This could be because the lecturers contributed to reducing the information asymmetry [4]. Although communication between administrative officers and students is an element of satisfaction in educational systems with reduced attendance [3], the role of lecturers compensated for this lack of communication. Lecturers were a key element in the process of adapting to the educational system imposed during the pandemic [5,18]. McKenzie et al. [52] found that establishing direct communication channels between lecturers and students helped maintain their trust. According to Schwanenberger et al. [14], lecturers serve as organizational and technological support during the learning process in online learning. These facts contributed to the maintenance of student satisfaction during the pandemic. These results are contrary to other studies on student satisfaction in online learning during the pandemic: Gallego-Gómez et al. [8] and Yang and Huang [20]—although nonorganizational variables were supported—found higher levels of student satisfaction in online learning.

The organizational models of online learning at UTEQ and UEB in 2020 were similar. The online model built was the same at each university analyzed; so, it can be extended to other universities [25]. However, according to the results of this study, the online model did not contribute to improving student satisfaction, but rather to maintaining it in an emergency situation. Therefore, a different strategy should be proposed to improve student satisfaction.

The RC model allows establishing strategies to improve communication [22] and increase satisfaction [3,23–25,27–29,50]. The combination of the best RC practices of face-to-face and online learning models in a hybrid model is proposed. As a concrete measure, the adaptation of the ICTs of the administrative officers is proposed. First of all, establishing a remote online appointment procedure can assure students that they will receive care within reasonable time frames. On the other hand, a technological improvement that allows multiple lines to be linked on the same device could optimize communication when administrative officers work from home. These measures could help improve communication flows and increase student satisfaction. These results agree with those of Schwanenberger et al. [14] and Harati et al. [13], who demonstrated that administrative officers need to improve their technological support to offer optimal communication to the student [19]. Furthermore, this could help free teachers from administrative functions and allow them to focus on their teaching tasks [1,3,12]. With more time available, lecturers could develop pedagogical skills that combine face-to-face learning and digital tools [19], such as gamification, which increases student motivation [4,12]. Synchronous and asynchronous methods could also be combined, such as video recording of face-to-face classes. Consequently, students could avoid the passive learning of online learning [32] and, at the same time, acquire a deeper understanding of the content through subsequent visualizations [19].

## 6. Limitations

The methodology used makes it possible to compare the two organizational models of HE: face-to-face and online. The creation of a hybrid system that integrates the best practices of both models is proposed. However, future research could validate the combination of specific practices and verify if the new system is associated with higher levels of satisfaction, since its exploration has not been contemplated in this research.

## 7. Conclusions

The relational coordination model was a useful tool to build an organizational model of online learning during the pandemic period. The organizational model of online learning at UTEQ changed, compared to face-to-face learning, even though the level of satisfaction was similar in both. The main changes observed were related to the cooperation and resolution of problems by the lecturers and administrative officers.

The role of lecturers gained importance beyond teaching tasks in times of confinement, also assuming administrative functions. However, the level of satisfaction was similar in both models since the lecturers compensated for the lack of communication between the students and the administrative officers to solve administrative problems. This organizational model is applicable to other universities in similar contexts. Nevertheless, since online learning was not associated with an improvement in satisfaction, different structural changes are necessary to improve it. The construction of a third educational model that combines the best practices of online and face-to-face learning in a hybrid system could be an alternative that improves student satisfaction.

**Supplementary Materials:** The following supporting information can be downloaded at: https://www.mdpi.com/article/10.3390/educsci12070448/s1, Relational coordination in higher education: Student survey; Table S1: Principal components (PC) loading matrix of rotated of UEB_2020; Table S2: Centroids for each cluster of UEB_2020; Figure S1: Statistical parameters of the satisfaction value for each university; Figure S2: Relational coordination clusters of UEB_2020; Figure S3: Satisfaction according to the first two components of UEB_2020; Figure S4: Total satisfaction in each relational coordination model of UEB_2020.

**Author Contributions:** Conceptualization and methodology, all authors. Formal analysis, software, data curation, data processing, C.C.-M. and Y.G.T.; statistical analysis, C.C.-M.; validation and investigation, C.D.-P.-H. and A.G.; supervision, A.G., C.D.-P.-H. and C.C.-M.; project administration, C.D.-P.-H. and A.G.; data acquisition, Y.G.T. and E.D.O. All authors have been involved in developing, writing, commenting, editing and reviewing the manuscript. All authors have read and agreed to the published version of the manuscript.

**Funding:** This research received no external funding.

**Institutional Review Board Statement:** Not applicable.

**Informed Consent Statement:** Not applicable.

**Data Availability Statement:** This is not applicable as the data are not in any data repository of public access; however, if editorial committee needs access, we will happily provide them, please use this email: pa1gamaa@uco.es.

**Conflicts of Interest:** The authors declare no conflict of interest.

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
