# Peer review of "An Organizational Model of Online Learning in the Pandemic Period: Comparison with Traditional Face-to-Face Learning"

_education, doi:10.3390/educsci12070448_

Round 1

Reviewer 1 Report

The topic of the paper is actual. The aim of the study is to present the organizational model of online learning built by the authors and implemented during the pandemic period and to compare it with previous face-to-face training. To achieve this goal, the authors define the appropriate research questions and seek their answers through a survey.
In the introduction, the authors discuss the literature related to communication in HE during the pandemic. This review can be extended if more examples are available.

The sections Materials and Methods and Results present the conducted research and the obtained results. They are illustrated with tables and diagrams and interpreted in details. In section 4.1.2, the authors mention Figure S1, which is not available in the article but in the supplementary materials. I recommend them to note this when they refer to the figure (not at the end of the article). 

Author Response

Response to Reviewer 1 Comments

Dear Education Sciences Journal Reviewer,

We appreciate the effort of your review and observations. We have taken them into account in the new version of the article. We think that your feedback has allowed us to improve the manuscript.

In the following points you can see how we have considered your recommendations:

Reviewer (R): The topic of the paper is actual. The aim of the study is to present the organizational model of online learning built by the authors and implemented during the pandemic period and to compare it with previous face-to-face training. To achieve this goal, the authors define the appropriate research questions and seek their answers through a survey. In the introduction, the authors discuss the literature related to communication in HE during the pandemic. This review can be extended if more examples are available.

Author (A): Based on this comment, we have expanded the literature review in section 1.1.

Old text:

During the confinement period, communication at the university was affected [1,5]. Given the need to limit physical contact, information and communication technologies (ICTs) have been a crucial element, so that digital literacy has been an essential element in online learning in the pandemic period [7,8]. Sosa Díaz et al. [4] and Van-Der-Velde et al. [9], studied student satisfaction in the online classroom, concluding that communication skills through ICTs are necessary in the learning process both for accessing to training content and to ensure optimal communication between the student and the other university worker’s profiles, mainly lecturers and administrative staff. This opinion is shared by Harati et al. [10] and Schwanenberger et al. [11], who observed it both in students and in university administrative officers in online teaching. Furthermore, Flores et al. [5] and Prieto-Ballester et al. [12], showed that the level of digital literacy for online communication during confinement directly and positively affected the student's well-being in online classes. In other words, HE requires quality organizational measures in the face of the change caused by Covid-19 [13,14]”.

New text (Lines 58-76):

“During the confinement period, communication at the university was affected [1,5]. Given the need to limit physical contact, information and communication technologies (ICTs) have been a crucial element, so that digital literacy has been an essential element in online learning in the pandemic period [7,8]. According to Tejedor et al. [9] and Simon et al. [10], the learning scenario established by the pandemic highlighted the need to improve the digital skills of university staff. In this context, communication was the key element to provide students with personalized information and encourage their participation [11]. Sosa Díaz et al. [4] and Van-Der-Velde et al. [12], studied student satisfaction in the online classroom, concluding that communication skills through ICTs are necessary in the learning process both for accessing to training con-tent and to ensure optimal communication between the student and the other university worker’s profiles, mainly lecturers and administrative staff. This opinion is shared by Harati et al. [13] and Schwanenberger et al. [14], who observed it both in students and in university administrative officers in online teaching. Obtaining timely feedback from lecturers in the evaluation process was crucial for students during online education in the pandemic period [15–17]. Furthermore, Flores et al. [5] and Prieto-Ballester et al. [18], showed that the level of digital literacy for online communication during confinement directly and positively affected the student's well-being in online classes. In other words, HE requires quality organizational measures in the face of the change caused by Covid-19 [15,19,20]”.

We have added the following works, which are added in the references section:

“9.       Tejedor, S.; Cervi, L.; Pérez-Escoda, A.; Jumbo, F.T. Digital Literacy and Higher Education during COVID-19 Lockdown: Spain, Italy, and Ecuador. Publications 2020, 8, 48, doi:10.3390/publications8040048.

  1. Simón, Y.; Grajales Melián, I.; Silva, C. Blended Learning for Doctoral Training in the Context of the COVID-19 Pandemic. Revista Tempos e Espaços em Educação 2022, 15, e16685, doi:10.20952/revtee.v15i34.16685.

  1. Istenič, A. Online Learning under COVID-19: Re-Examining the Prominence of Video-Based and Text-Based Feedback. Educ Technol Res Dev 2021, 69, 117–121, doi:10.1007/s11423-021-09955-w.

  1. Lovell, D.; Dolamore, S.; Collins, H. Examen de Los Desajustes de Comunicación de Las Organizaciones Públicas Durante El COVID-19 a Través de La Lente de La Educación Superior. Administration & Society 2022, 54, 212–247, doi:10.1177/00953997211026949.

  1. Almossa, S.Y. University Students’ Perspectives toward Learning and Assessment during COVID-19. Educ Inf Technol 2021, 26, 7163–7181, doi:10.1007/s10639-021-10554-8.

  1. Zikargae, M.M.H. Risk Communication, Ethics and Academic Integrity in the Process of Minimizing the Impacts of the Covid-19 Crisis in Ethiopian Higher Education. Cogent Education 2022, 9, 2062892, doi:10.1080/2331186X.2022.2062892”.

Due to the addition of these references, the rest of them, since the ninth, have been renumbered.

R: Line 353.The sections Materials and Methods and Results present the conducted research and the obtained results. They are illustrated with tables and diagrams and interpreted in details. In section 4.1.2, the authors mention Figure S1, which is not available in the article but in the supplementary materials. I recommend them to note this when they refer to the figure (not at the end of the article).

A: Line 353. The writing of this sentence has been modified to clarify the existence of the supplementary material, according to the comment.

Old text:

“The satisfaction indicator showed differences between each group regarding its density function (Figure S1)”.

New text (373-374):

“The satisfaction indicator showed differences between each group regarding its density function. It was shown in Figure S1 as supplementary material”.

Finally, a general correction of the English language has been made. In addition, we have clarified the wording of sections 3 and 4 (materials and methods and results) to make it easier to understand.

Reviewer 2 Report

Very interesting and reevelant manuscript

Author Response

Dear Education Sciences Journal Reviewer,

We appreciate the effort of your review and comments. We have taken them into account in the new version of the article. We have reviewed the English language and have made a general spelling correction. In addition, we have expanded the theoretical framework of our research in section 1.1. and we have clarified the wording of sections 3 and 4 (materials and methods and results). We believe that your comments have enabled us to improve the manuscript and we are pleased that you find our manuscript an interesting and insightful contribution.

Reviewer 3 Report

This article addresses the proposal of an organizational model for online learning in higher education institutions, which was made by producing and comparing data collected in two Ecuadorian universities during the pandemic period with previous pre-pandemic data collected in one of the two universities considered. The introduction is well written and mentions appropriate references. It also presents the objective and the research questions. Then the context in which the study was carried out, in Ecuadorian universities, is described. In general, the results are well described, and limitations of the study and some conclusions are presented, which are appropriate.

My biggest concern with this paper is section 3., on Materials and Methods, and I propose that it be revised considerably. In my opinion, the section is a bit confusing. It starts with a short presentation of the methodological approach followed immediately by a description of the participants and the sample of data obtained, namely those resulting from the first questions of the questionnaire (section 3.1), which is only presented in the next section (3.2). Next, in section 3.2, the structure of the questionnaire is presented, as well as other aspects, including the Likert scale used. Elements on the outcome of the application of the questionnaire, namely its reliability, are also presented. Section 3.3 describes the set of techniques used to process the data, again followed by some information that results from its application. Thus, I consider that:

- Section 3. should objectively describe the methods and techniques used, and just that. That is, it should allow the reader to understand how the research was planned and carried out, its phases, the design and application of the instruments, the participants in the study, and the data processing techniques. A figure describing the research would be welcome.

- All the results should be in section 4, which should begin by describing the samples obtained (namely the data in Table 1), followed by all the other results.

- I believe that clarifying well this separation between the Methods and techniques used, in section 3, and the results, in section 4, will make the paper more readable.

A few more observations that the author should consider:

- In Table 1 he puts information about p-value: why? In relation to this figure and the text that precedes it, the information about the courses is not clear: 1st, 2nd, etc, they refer to which courses?

- Table 2 presents the questions of the questionnaire, which are in English. Given the location of the universities where the study was carried out, was the original questionnaire in Spanish? If so, it should be mentioned that what is in Table 2 is a translation of the original, which was in Spanish.

- In line 196 it mentions that a Likert scale was used, "from 1 (uncommon) to 5 (very common)". Was this scale used for all the questions? Why was it used? I don't think this scale is appropriate for several questions. For example, one of the most obvious is the case of the Satisfaction questions: why a satisfaction scale wasn’t used, something like 1 (totally dissatisfied) to 5 (totally satisfied). In the case of the scale used in the study, it would also be important to know the wording of all the elements of the scale

I wish the authors all the best for this work and their future work

Author Response

Response to Reviewer 3 Comments

Dear Education Sciences Journal Reviewer,

We appreciate the effort of your review and observations. We have taken them into account in the new version of the article. We think that your feedback has allowed us to improve the manuscript.

In the following points you can see how we have considered your recommendations:

Reviewer (R): This article addresses the proposal of an organizational model for online learning in higher education institutions, which was made by producing and comparing data collected in two Ecuadorian universities during the pandemic period with previous pre-pandemic data collected in one of the two universities considered. The introduction is well written and mentions appropriate references. It also presents the objective and the research questions. Then the context in which the study was carried out, in Ecuadorian universities, is described. In general, the results are well described, and limitations of the study and some conclusions are presented, which are appropriate.

My biggest concern with this paper is section 3., on Materials and Methods, and I propose that it be revised considerably. In my opinion, the section is a bit confusing. It starts with a short presentation of the methodological approach followed immediately by a description of the participants and the sample of data obtained, namely those resulting from the first questions of the questionnaire (section 3.1), which is only presented in the next section (3.2). Next, in section 3.2, the structure of the questionnaire is presented, as well as other aspects, including the Likert scale used. Elements on the outcome of the application of the questionnaire, namely its reliability, are also presented. Section 3.3 describes the set of techniques used to process the data, again followed by some information that results from its application.

Author (A): Based on your comments, we have made changes throughout the text, especially in the materials and methods and results sections, to clarify the text.

R: Thus, I consider that:

- Section 3. should objectively describe the methods and techniques used, and just that. That is, it should allow the reader to understand how the research was planned and carried out, its phases, the design and application of the instruments, the participants in the study, and the data processing techniques. A figure describing the research would be welcome.

- All the results should be in section 4, which should begin by describing the samples obtained (namely the data in Table 1), followed by all the other results.

- I believe that clarifying well this separation between the Methods and techniques used, in section 3, and the results, in section 4, will make the paper more readable.

A: To clarify sections 3 and 4 the following modifications have been made:

Only information relating to the methods and techniques used has been maintained. Therefore, the sociodemographic description of the sample has been removed from this section and has been inserted at the beginning of section 4 of results, based on your comments.

Thus, the old section 3.1. was displayed like this:

“3.1. HE in pandemic period: Online learning at UTEQ_2020 and UEB_2020

The participants were students enrolled in undergraduate and postgraduate degrees in the fields of knowledge: social sciences, humanities, engineering and health sciences. More than 3,000 surveys were collected between the months of May and September in 2020 at UTEQ and UEB. The surveys were distributed online through the Google Forms application. The answers of the students that were incomplete and those in which logical inconsistencies were observed were eliminated in the database cleaning [39]. A stratified random sample composed of 1,147 from online learning (UTEQ_2020) was used to determinate the main model. Finally, another stratified random sample and composed of 1,627 from online learning (UEB_2020) was used to validate the model of online learning (UTEQ_2020).

The main sociodemographic data of each sample are shown in Table 1. The majority of students surveyed in 2020 were female and younger than 25 years old. First and second year students and postgraduate students predominated in UTEQ_2020 and UEB_2020. In the 2020 samples, students from all university courses were considered, since the online modality during confinement affected all students. Most of the online learning (UTEQ_2020) students belonged to the engineering field of knowledge, due to the predominantly agrarian nature of the university. The distribution of the sample by field of knowledge in face-to-face learning (UTEQ_2018) confirmed this point. In UEB_2020, most of the students belonged to the social sciences, engineering and health sciences for the most part, which confirmed the heterogeneity of the sample. The de-scription of the UTEQ_2018 sample is derived from the work of Checa et al. [23].

Table 1. Sociodemographic distribution of the samples (%)”.

It has been modified and this is the new text (Lines 189-200):

“3.1. HE in pandemic period: Online learning at UTEQ_2020 and UEB_2020

The participants were students enrolled in undergraduate and postgraduate degrees in the fields of knowledge: social sciences, humanities, engineering and health sciences. More than 3,000 surveys were collected between the months of May and September in 2020 at UTEQ and UEB. The surveys were distributed online through the Google Forms application. The answers of the students that were incomplete and those in which logical inconsistencies were observed were eliminated in the database cleaning [39]. A stratified random sample composed of 1,147 from online learning (UTEQ_2020) was used to determinate the main model. Finally, another stratified random sample and composed of 1,627 from online learning (UEB_2020) was used to validate the model of online learning (UTEQ_2020)”.

Besides, the old section 4 output was displayed like this:

“4. Results

4.1. Organizational model in online learning (UTEQ_2020)

4.1.1. Organizational typology of online learning (UTEQ_2020)”

It has been modified and this is the new text (Lines 304-318):

“4. Results

The main sociodemographic data of each sample are shown in Table 2. The majority of students surveyed in 2020 were female and younger than 25 years old. First and second year students and postgraduate students predominated in UTEQ_2020 and UEB_2020. In the 2020 samples, students from all university courses were considered, since the online modality during confinement affected all students. Most of the online learning (UTEQ_2020) students belonged to the engineering field of knowledge, due to the predominantly agrarian nature of the university. The distribution of the sample by field of knowledge in face-to-face learning (UTEQ_2018) confirmed this point. In UEB_2020, most of the students belonged to the social sciences, engineering and health sciences for the most part, which confirmed the heterogeneity of the sample. The de-scription of the UTEQ_2018 sample is derived from the work of Checa et al. [23].

Table 2. Sociodemographic distribution of the samples (%)”.

It is important to note that tables 1 and 2 have exchanged their numbering due to the change in position in the text.

In addition, according to your suggestion, a figure has been added that explains the process of this investigation. It appears in the text on line 186. The figure is as follows:

Figure 1. Methodology stages.

The text of section 3, where the three stages of the methodology are explained, has been expanded to indicate the introduction of this new figure:

Old text:

“In a first stage, an organizational model was built for online learning (UTEQ_2020), through the development of an organizational typology. Principal com-ponent analysis (PCA) and cluster analysis were performed. Subsequently, the model obtained was related to student satisfaction through the use of generalized linear mod-els (GLM). In a second stage, to deepen in the organizational differences between face-to-face learning (UTEQ_2018) and online learning (UTEQ_2020), both models were compared. In a third stage, the online organizational learning (UTEQ_2020) was vali-dated with data from the online learning model of the State University of Bolivar (UEB_2020)”.

New text (Lines 176-185):

  1. Materials and Methods

“In a first stage, an organizational model was built for online learning (UTEQ_2020), through the development of an organizational typology. Principal com-ponent analysis (PCA) and cluster analysis were performed. Subsequently, the model obtained was related to student satisfaction through the use of generalized linear mod-els (GLM). In a second stage, to deepen in the organizational differences between face-to-face learning (UTEQ_2018) and online learning (UTEQ_2020), both models were compared. In a third stage, the online organizational learning (UTEQ_2020) was vali-dated with data from the online learning model of the State University of Bolivar (UEB_2020). The methodological stages are shown in Figure 1”.

It's important pointing that, when introducing this figure, the rest of the figures in the text have been renumbered to give concordance to the wording.

Finally, to improve the methodology section, a fragment has been added in section 3.3., regarding the sociodemographic descriptive analysis of the sample.

The old text:

“3.3. Statistical analysis

To answer RQ1, an organizational model was developed for online learning (UTEQ_2020) [42,43]”.

In has been modified (Line 249):

“3.3. Statistical analysis

Firstly, a descriptive statistical analysis was performed to observe the sociodemographic distribution of the sample.

Subsequently, to answer RQ1, an organizational model was developed for online learning (UTEQ_2020) [42,43]”.

R: A few more observations that the author should consider:

- In Table 1 he puts information about p-value: why? In relation to this figure and the text that precedes it, the information about the courses is not clear: 1st, 2nd, etc, they refer to which courses?

A: Considering the comments, we have eliminated the p-value columns, since what this table exposes are a merely descriptive analysis and this indicator was not appropriate in this case.

On the other hand, in the new Table 2, when we wrote "course" we meant "academic year". We have fixed this bug (New Table 2, which before the modification was Table 1).

It has been clarified in the table as follows:

The original text in the table: “Course: “1º, 2º; 3º; 4º, 5º, 6º,7º” has been modified by “Academic year: 1st, 2nd, 3rd, 4th,5th,6th,7th”.

Thus, the new Table 2 is finally displayed in this way:

“Table 2. Sociodemographic distribution of the samples (%).

Age

Gender

Academic year

Field of knowledge

< 25

> 25

Male

Female

1st

2nd

3rd

4th

5th

6th

7th

Social sciences

Humanities

Engineering

Health sciences

UTEQ_2018

3,200

43.36

56.64

38.71

61.29

-

-

45.94

42.98

11.08

-

-

37.25

55.36

7.39

UTEQ_2020

1,147

88.40

11.60

40.71

59.29

19.70

13.51

10.46

12.29

13.86

7.67

22.49

-

-

100

-

UEB_2020

1,627

83.90

16.10

40.69

59.31

1.60

18.87

17.95

17.15

16.04

9.04

19.36

37.19

5.84

29.19

27.78

R: - Table 2 presents the questions of the questionnaire, which are in English. Given the location of the universities where the study was carried out, was the original questionnaire in Spanish? If so, it should be mentioned that what is in Table 2 is a translation of the original, which was in Spanish.

A: Considering the comments, this point has been clarified within the text. In addition, we remind that due to the previous change, the old Table 2 becomes Table 1 in this new version of the text:

Old text:

“[…] 6 items on satisfaction [21–23,25] (Table 2). The questions in the survey were […]”.

New text (Lines 215-217):

“[…] 6 items on satisfaction [21–23,25]. The original survey questions were asked in Spanish and are shown in an English translation in Table 1. The questions in the survey were […]”.

R: - In line 196 it mentions that a Likert scale was used, "from 1 (uncommon) to 5 (very common)". Was this scale used for all the questions? Why was it used? I don't think this scale is appropriate for several questions. For example, one of the most obvious is the case of the Satisfaction questions: why a satisfaction scale wasn’t used, something like 1 (totally dissatisfied) to 5 (totally satisfied). In the case of the scale used in the study, it would also be important to know the wording of all the elements of the scale.

A: To clarify this point, the text was modified as follows:

Old text:

“A Likert scale metric was used, from 1 (uncommon) to 5 (very common). The intervals between the points of the scale corresponded to empirical observations in the metric sense. A visual analog scale was displayed for each survey question presented to the students [40]”.

New text (Lines 220-222):

“In order to answer the survey questions obtaining metric variables, a Likert scale from 1 to 5 was used. Each equidistant point on the visual scale was associated with a level of student response [40]”.

In order to show the wording of all the elements of the scale, the complete survey has been added as supplementary material.

Old text:

“The complete showed a Cronbach's alpha of 0.957 [41]”.

New text (Lines 228-229):

“The complete survey (available as supplemental material) showed a Cronbach's alpha of 0.957 [41]”.

Old text:

Supplementary Materials: The following supporting information can be downloaded at: www.mdpi.com/xxx/s1, Table S1: Principal components (PC) loading matrix of rotated of UEB_2020; Table S2: Centroids for each cluster of UEB_2020; Figure S1: Statistical parameters of the satisfaction value for each university; Figure S2: Relational coordination clusters of UEB_2020; Figure S3: Satisfaction according to the first two components of UEB_2020; Figure S4: Total satisfaction in each relational coordination model of UEB_2020”.

New text (Lines 513-519):

Supplementary Materials: The following supporting information can be downloaded at: www.mdpi.com/xxx/s1, Relational coordination in higher education: Student survey; Table S1: Principal components (PC) loading matrix of rotated of UEB_2020; Table S2: Centroids for each cluster of UEB_2020; Figure S1: Statistical parameters of the satisfaction value for each university; Figure S2: Relational coordination clusters of UEB_2020; Figure S3: Satisfaction according to the first two components of UEB_2020; Figure S4: Total satisfaction in each relational coordination model of UEB_2020”.

R: Is the content succinctly described and contextualized with respect to previous and present theoretical background and empirical research (if applicable) on the topic? – Must be improved.

A: Finally, we have made a general correction of the English language and an extension of the theoretical framework in section 1.1.

Old text:

During the confinement period, communication at the university was affected [1,5]. Given the need to limit physical contact, information and communication technologies (ICTs) have been a crucial element, so that digital literacy has been an essential element in online learning in the pandemic period [7,8]. Sosa Díaz et al. [4] and Van-Der-Velde et al. [9], studied student satisfaction in the online classroom, concluding that communication skills through ICTs are necessary in the learning process both for accessing to training content and to ensure optimal communication between the student and the other university worker’s profiles, mainly lecturers and administrative staff. This opinion is shared by Harati et al. [10] and Schwanenberger et al. [11], who observed it both in students and in university administrative officers in online teaching. Furthermore, Flores et al. [5] and Prieto-Ballester et al. [12], showed that the level of digital literacy for online communication during confinement directly and positively affected the student's well-being in online classes. In other words, HE requires quality organizational measures in the face of the change caused by Covid-19 [13,14]”.

New text (Lines 58-76):

“During the confinement period, communication at the university was affected [1,5]. Given the need to limit physical contact, information and communication technologies (ICTs) have been a crucial element, so that digital literacy has been an essential element in online learning in the pandemic period [7,8]. According to Tejedor et al. [9] and Simon et al. [10], the learning scenario established by the pandemic highlighted the need to improve the digital skills of university staff. In this context, communication was the key element to provide students with personalized information and encourage their participation [11]. Sosa Díaz et al. [4] and Van-Der-Velde et al. [12], studied student satisfaction in the online classroom, concluding that communication skills through ICTs are necessary in the learning process both for accessing to training con-tent and to ensure optimal communication between the student and the other university worker’s profiles, mainly lecturers and administrative staff. This opinion is shared by Harati et al. [13] and Schwanenberger et al. [14], who observed it both in students and in university administrative officers in online teaching. Obtaining timely feedback from lecturers in the evaluation process was crucial for students during online education in the pandemic period [15–17]. Furthermore, Flores et al. [5] and Prieto-Ballester et al. [18], showed that the level of digital literacy for online communication during confinement directly and positively affected the student's well-being in online classes. In other words, HE requires quality organizational measures in the face of the change caused by Covid-19 [15,19,20]”.

We have added the following works, which are added in the references section (Lines 559-564; 573-580):

“9.          Tejedor, S.; Cervi, L.; Pérez-Escoda, A.; Jumbo, F.T. Digital Literacy and Higher Education during COVID-19 Lockdown: Spain, Italy, and Ecuador. Publications 2020, 8, 48, doi:10.3390/publications8040048.

  1. Simón, Y.; Grajales Melián, I.; Silva, C. Blended Learning for Doctoral Training in the Context of the COVID-19 Pandemic. Revista Tempos e Espaços em Educação 2022, 15, e16685, doi:10.20952/revtee.v15i34.16685.

  1. Istenič, A. Online Learning under COVID-19: Re-Examining the Prominence of Video-Based and Text-Based Feedback. Educ Technol Res Dev 2021, 69, 117–121, doi:10.1007/s11423-021-09955-w.

  1. Lovell, D.; Dolamore, S.; Collins, H. Examen de Los Desajustes de Comunicación de Las Organizaciones Públicas Durante El COVID-19 a Través de La Lente de La Educación Superior. Administration & Society 2022, 54, 212–247, doi:10.1177/00953997211026949.

  1. Almossa, S.Y. University Students’ Perspectives toward Learning and Assessment during COVID-19. Educ Inf Technol 2021, 26, 7163–7181, doi:10.1007/s10639-021-10554-8.

  1. Zikargae, M.M.H. Risk Communication, Ethics and Academic Integrity in the Process of Minimizing the Impacts of the Covid-19 Crisis in Ethiopian Higher Education. Cogent Education 2022, 9, 2062892, doi:10.1080/2331186X.2022.2062892”.

Due to the addition of these references, the rest of them, since the ninth, have been renumbered.
